# Responses of marine trophic levels to the combined effects of ocean acidification and warming

Nan Hu [1], Paul E. Bourdeau[2] & Johan Hollander [3] ✉

Marine organisms are simultaneously exposed to anthropogenic stressors associated with ocean acidification and ocean warming, with expected interactive effects. Species from different trophic levels with dissimilar characteristics and evolutionary histories are likely to respond differently. Here, we perform a meta-analysis of controlled experiments including both ocean acidification and ocean warming factors to investigate single and interactive effects of these stressors on marine species. Contrary to expectations, we find that synergistic interactions are less common (16%) than additive (40%) and antagonistic (44%) interactions overall and their proportion decreases with increasing trophic level. Predators are the most tolerant trophic level to both individual and combined effects. For interactive effects, calcifying and non-calcifying species show similar patterns. We also identify climate region-specific patterns, with interactive effects ranging from synergistic in temperate regions to compensatory in subtropical regions, to positive in tropical regions. Our findings improve understanding of how ocean warming, and acidification affect marine trophic levels and highlight the need for deeper consideration of multiple stressors in conservation efforts.

Global climate change, characterized by unprecedented rapidity, substantially impacts marine ecosystems[1–5]. By the year of 2100, global ocean surface temperatures are predicted to increase by about 4 °C compared with the 1990s under the RCP 8.5 scenario (business-as-usual), and together with elevated $CO_2$ levels, will additionally lead to a reduction in pH by 0.3–0.5 units (ocean acidification)[6–8]. These shifts in temperature (ocean warming, hereafter referred to as OW) and ocean carbonate chemistry (ocean acidification, hereafter OA) are considered two of the greatest threats to marine organisms[2], from primary producers at the base of food webs on through to intermediate consumers and top-predators[3,9].

OA and OW's effects are well-documented, especially the lowered calcium carbonate saturation state caused by OA, which negatively impacts a range of calcifying marine organisms[10–13] but see ref. 14. Potential consequences include community-level imbalances favoring

non-calcifying organisms and food-web simplification[15]. In particular, OA can lead to higher costs associated with protein synthesis and turnover, and the maintenance of the intercellular acid-base balance required for a series of physiological functions across a wide range of species[16,17]. In contrast, some autotrophs may actually benefit from OA, since a lower pH can increase the availability of substrates used for photosynthesis and/or decrease the energy needed for carbon-concentrating mechanisms[18,19]. Previous studies have also suggested that OW can lead to increased metabolic rates[20] and increased consumption rates[21] with shifts in food-web structures[20,22].

Focusing solely on single-stressor effects overlooks climate change's multifaceted nature. As investigators increasingly recognize, single-stressor studies may inaccurately represent marine global climate change effects[23,24]. Interactions among multiple climatic stressors, where the ecological effect of one stressor is dependent on the

[1]Department of Biology- Aquatic Ecology, Lund University, Lund, Sweden. [2]Department of Biological Sciences, California State Polytechnic University, Humboldt, Arcata, CA, USA. [3]World Maritime University, Ocean Sustainability, Governance & Management Unit, 211 18 Malmö, Sweden. ✉e-mail: johan.hollander@wmu.se

magnitude of another stressor, are very common across ecosystems[25–27]. Such dynamics suggest that it is important to investigate organismal responses not only to single stressors but also to the interacting stressors that species will experience. For instance, multiple stressors can lead to non-additive interaction outcomes, where the combined effects are more or less than expected from additive models, such as synergistic or antagonistic effects, respectively[25,28]. For example, for marine ecosystems, temperature can influence seawater carbonate chemistry, such that OW will affect OA by decreasing $CO_2$ solubility and affecting the dissociation rate of the carbonate system, leading to higher saturation state with the same $CO_2$ concentration[29]. Accordingly, it is essential to consider all ramifications of global change, specifically, since there are reports of adverse impacts on marine organisms attributed to global warming[25]. Conversely, alternative studies have revealed that elevated seawater temperatures can serve as a mitigating factor against ocean acidification[25,30]. These effects may also exert differential influences on species across various climate regions (e.g., Tropical, Subtropical, and Temperate). A central question regarding interactive effects is thus whether in tropical regions, species residing near their upper temperature limits (temperatures above 40 °C may disrupt physiological functions), or tropical species, adapted to higher temperatures, are less affected by heat stress and may find relief from ocean warming offsetting low pH effects. While in temperate regions, warmer temperatures may benefit temperate calcifying species by compensating for ocean acidification – conversely, temperate species, accustomed to colder waters, can be significantly impacted by even a small temperature increase, highlighting the importance of relative temperature changes in assessing organismal vulnerability. Therefore, it is imperative to refrain from preconceiving that global change invariably leads to negative outcomes, especially in cases of cumulative effects, where one stressor may mask the impact of another (antagonistic interactions)[27,31]. In this case and to follow the terminology in the field – here, we refer to a stressor as any natural or anthropogenic factor that leads to a measurable alteration in biological reactions, whether the change is favorable or adverse; similar to a 'driver'[32,33].

Furthermore, dynamic shifts in climatic stressors are complex and vary on diurnal and seasonal timescales across a wide range of marine ecosystems[34–37]. It is therefore essential to investigate responses to interacting stressors. As seen in tide pools and coral reefs, global stressors can vary diurnally and seasonally, leading to potentially synergistic or antagonistic effects[38,39]. Therefore, investigating these effects across various trophic levels becomes an essential first step to understand the complexities of marine ecosystem interactions. Previous studies have shown that different trophic levels vary widely in their sensitivity to climatic stressors across ecosystems[9,22,40,41]. Specifically, with respect to marine ecosystems, herbivores that are primary consumers and represent a low position in the food web are the most vulnerable group to OA and OW, whereas higher trophic levels (i.e., predators) demonstrate greater tolerances than lower trophic levels[9,41]. Although environmental stress models that have been influential in marine ecology suggest that environmental stress effects should vary predictably across trophic levels[42,43], a basic understanding whether multiple stressor effects vary across marine trophic levels remains unknown. In the present study, we assemble a large dataset including 486 observations from 162 fully factorial experiments in a meta-analysis to examine whether marine species from different trophic levels demonstrate differential responses to the single and combined stress of OA and OW (Fig. 1). We also tested if the distribution of interaction types (i.e., additive, synergistic, and antagonistic) differs across trophic levels. Finally, we additionally assessed how stressors, individually and in combination, influence marine species along a latitudinal gradient and among climate regions.

## Results

### Effects of ocean acidification, ocean warming, and their interaction on marine trophic levels

Overall, OA, OW, and their combination affected marine species differently ($Q_M = 15.39$, $p = 0.002$; Supplementary Table 3). Our data revealed that both OA (LnRR = −0.110, 95% CI = −0.185 to −0.034) and OW (LnRR = −0.106, 95% CI = −0.182 to −0.030) exerted detrimental effects, although the combined effect remained negligible (Fig. 2; Supplementary Table 4). Antagonistic and additive interactions accounted for 44% and 40% of all interactions, respectively, with synergistic interactions being minimal at 16% (Fig. 2).

Trophic levels responded differently to these stressors ($Q_M = 22.28$, $p = 0.035$; Supplementary Table 3). Primary producers benefited from OA, while meso-predators and top-predators remained unaffected. Combining meso-predators and top-predators (Predator) resulted in a marginal negative OA effect (Fig. 2; Supplementary Table 4). In contrast, herbivores suffered significant negative effects from OA (Fig. 2; Supplementary Table 4). OW exhibited similar effects across trophic levels, positively impacting primary producers and predators, but negatively affecting herbivores. We noted similar patterns for OA and OW across marine trophic levels, with detrimental effects decreasing as trophic rank increased (Fig. 2; Supplementary Table 4). However, the combined effect, while following a similar pattern, remained insignificant, suggesting additive interactions across trophic levels (Fig. 2; Supplementary Table 4).

Interaction types varied significantly across different trophic levels ($\chi^2 = 14.24$, $p = 0.027$, $df = 6$, $n = 162$). Additive effects were predominantly observed across trophic levels, increasing with the rank of trophic levels and irrespective of whether meso-predators and top-predators were merged or not (Donut plots in Fig. 2), while synergistic effects were less common (<17%). Synergistic interactions were less common for predators in general, and were absent in top-predators (Fig. 2).

### Effects of ocean acidification, ocean warming, and their interaction on calcifying and non-calcifying species at different trophic levels

Calcifiers and non-calcifiers were affected differently by stressors ($Q_M = 12.29$, $p = 0.007$; Supplementary Table 3). OA negatively affected calcifiers, but OW and the combination of OA and OW had no effect (Fig. 3a; Supplementary Table 5). Herbivores were significantly adversely affected by OA, while primary producers and predators were minimally affected (Fig. 3a; Supplementary Table 5). OW positively affected primary producers and predators, while herbivores experienced a negative effect of OW (Fig. 3a). The combined effects of OA and OW were non-significant across all trophic levels, with dominant additive and antagonistic interactions in calcifying species, and a minimal occurrence of synergistic effects (<20%) (Fig. 3a); however, the frequencies of interaction types did not differ significantly ($\chi^2 = 5.54$, $p = 0.477$, $df = 6$, $n = 113$).

Stressors had different effects on non-calcifying species ($Q_M = 9.38$, $p = 0.025$; Supplementary Table 3). OW had significant negative effects, while OA and combined effects were insignificant (Fig. 3b; Supplementary Table 6). OA positively affected non-calcifying primary producers, but negatively affected herbivores and meso-predators (Fig. 3b; Supplementary Table 6). OW had a large mean positive effect on top-predators (lnRR = 0.206 corresponding to a 22.8% change), yet a high degree of variability made this effect non-significant (Fig. 3b; Supplementary Table 6). Overall, we observed significant variation in interaction types among non-calcifying species across different trophic levels ($\chi^2 = 12.74$, $p = 0.047$, $df = 6$, $n = 49$), with synergistic interactions being less common than additive and antagonistic effects (Fig. 3b).

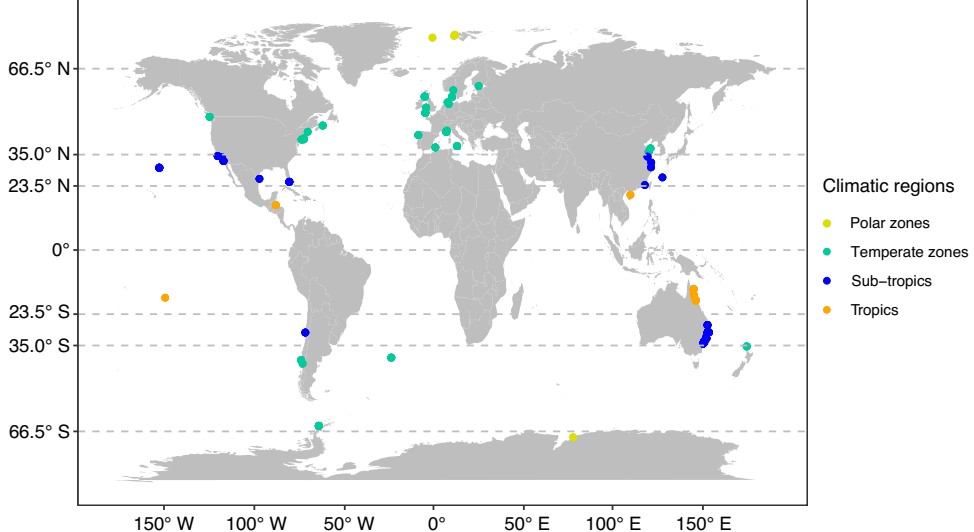

**Fig. 1 | Global distribution of factorial experimental climate change studies of ocean acidification and ocean warming on marine species.** Climatic regions are denoted by points color.

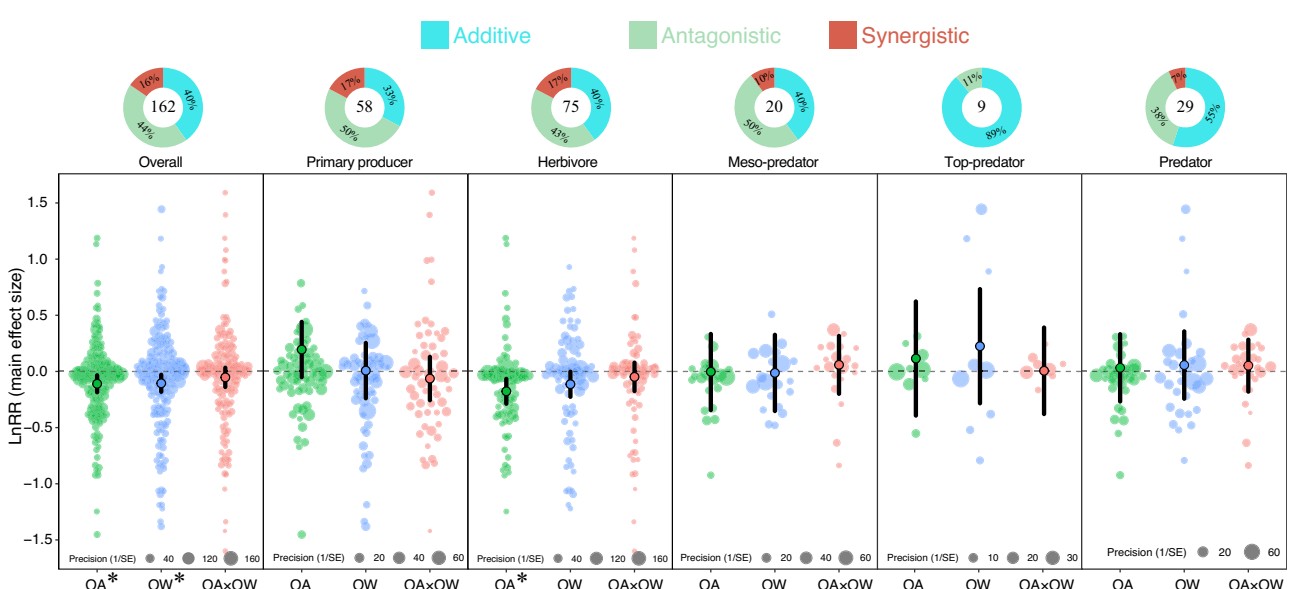

**Fig. 2 | Orchard plots showing mean of main effect size, confidence interval (CIs, bold line), and individual effect size with precision (1/SE) for ocean acidification (green), ocean warming (blue), and their combined effects (red) on marine trophic levels.** Mean effect size and 95% confidence intervals were estimated from multi-level meta-analytic models (two-sided) included trophic levels and stressors as moderators using main effect sizes. 95% confidence interval does not overlap with zero indicating significant effect showing by asterisk ($0.01 < *p < 0.05$; $0.001 < **p < 0.01$; $***p < 0.001$). The panel of Predator was formed by merging the Meso-predator and top-predator. Donut charts indicate the frequencies (%) of additive, antagonistic, and synergistic interaction types. Numbers inside donut charts indicate the number of observations ($k$).

## Relationship between main effect sizes and latitude across trophic levels

Linear relationships between the main effect sizes and latitudes indicated varying trends among trophic levels, though most trends were statistically non-significant (Fig. 4; Supplementary Table 7). In low-latitude species, a negative slope indicated a higher tolerance to stressors. On the other hand, a positive slope indicated an increased sensitivity in tropical species when compared to those in higher latitudes. Furthermore, the degree of variation between tropical and polar regions was represented by the steepness of this slope. A steeper slope implies a more pronounced difference between these regions. OA consistently affected all species along the latitudinal gradient, except top-predators, which showed a significant negative slope. However, we consider this a weak result due to a small sample size ($n = 9$) (Fig. 4). In

contrast, the significant negative intercept of herbivores, reflects that herbivores in low latitudes were significantly negatively affected by OA (Fig. 4). OW showed no significant latitude-related effects on any trophic levels, except primary producers. Interestingly, the combined effects resembled those of individual OA effects, with some deviations for herbivores (Fig. 4).

## Effects of ocean acidification, ocean warming, and their interaction on marine species across different climate regions

Overall, the individual effects of OA, OW, and their combined impact on tropical species were not significantly different ($Q_M = 7.12$, $p = 0.068$; Supplementary Table 3). The individual effects of OA and OW were similar and negligible, yet their combined effects were notably positive (Fig. 5; Supplementary Table 8). On the other hand, for

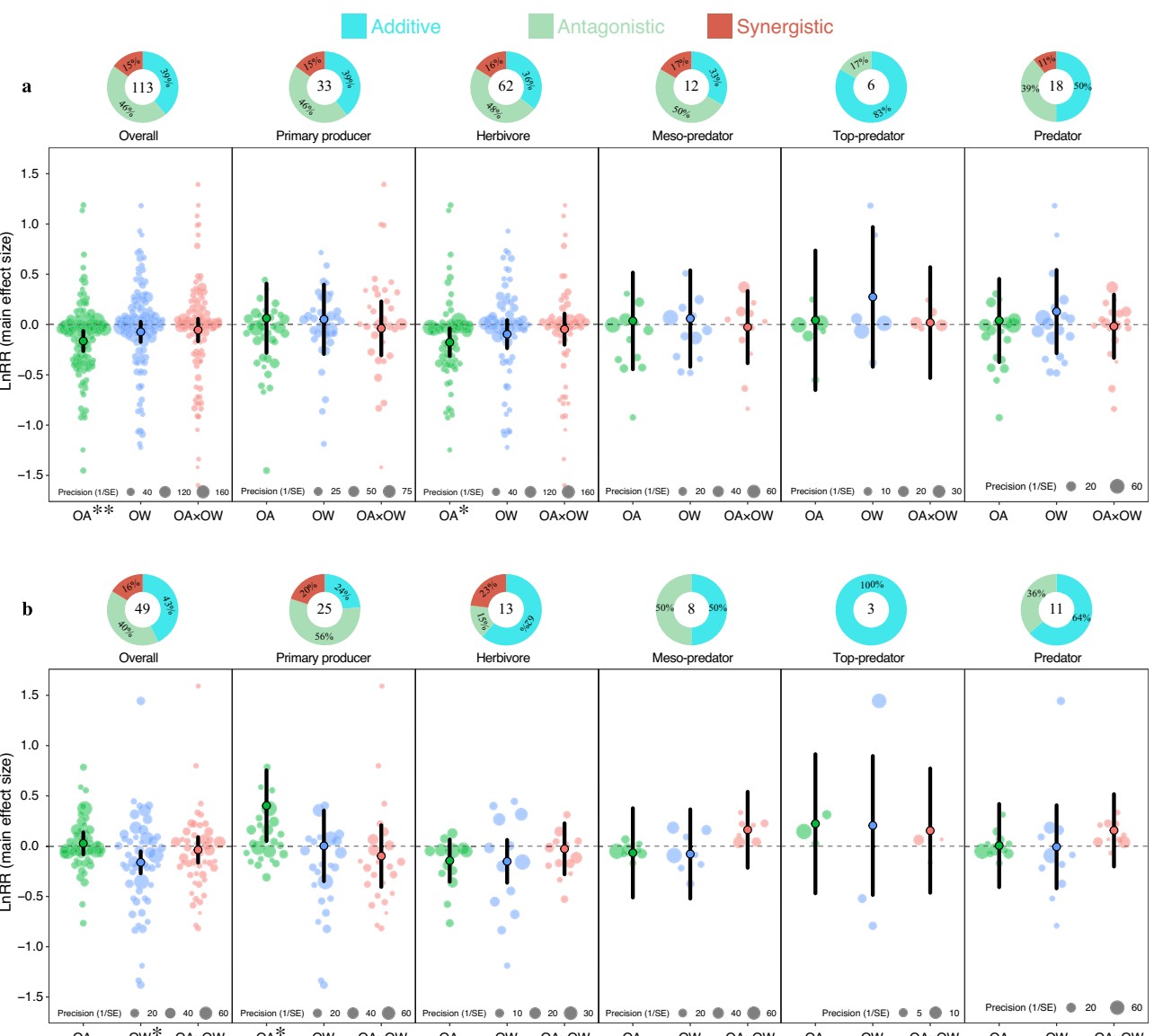

**Fig. 3 | Climatic effects on marine calcifiers and non-calcifiers.** Orchard plots showing mean of main effect size, confidence interval (CIs, bold line) and individual effect size with precision (1/SE) for ocean acidification (green), ocean warming (blue) and their combined effects (red) on marine trophic levels of calcifiers (**a**) and non-calcifiers (**b**). Mean effect size and 95% confidence intervals were estimated from multi-level meta-analytic models (two-sided) included trophic levels and stressors as moderators using main effect sizes. 95% confidence interval does not overlap with zero indicating significant effect showing by asterisk (0.01 < \*p < 0.05; 0.001 < \*\*p < 0.01; \*\*\*p < 0.001). The panel of Predator was formed by merging the Meso-predator and top-predator. Donut charts indicate the frequencies (%) of additive, antagonistic and synergistic interaction types. Numbers inside donut charts indicate the number of observations (k).

subtropical species OA and OW individually had marked negative effects ($Q_M$ = 13.98, p = 0.003; Supplementary Table 3), but in combination had no effect (Fig. 5; Supplementary Table 8). For temperate species, OA and OW individually showed minor negative effects ($Q_M$ = 7.44, p = 0.059; Supplementary Table 3), but their combined effect was significantly detrimental.

Interaction effect types differed among climate regions ($\chi^2$ = 26.57, p < 0.001, df = 4, n = 162; Fig. 5). The majority of effects were additive and antagonistic across these regions. Synergistic interactions were relatively rare, being completely absent in tropical regions, while accounting for 23% in subtropical regions, and 13% in temperate regions (Fig. 5).

## Discussion

Our results revealed that herbivores were the most sensitive trophic level compared to predators and primary producers, which, in contrast, were highly tolerant to OA and OW stressors and their interactive effects. These results support previous findings of a recent meta-analysis[9], despite the current study using only one-third of the number of species and studies of the previous analysis. The main reason for the smaller number of studies included here was because this meta-analysis only included fully factorial designs with both ocean warming and acidification. Although the combined effects of OA and OW did not vary among trophic levels, the pattern of mean effect sizes was however once again consistent with previous research about marine trophic levels and tolerance against climate change for individual stressors. For example, Hu et al.[9,41] demonstrated a pattern, similar to the results presented here, whereby the effects of OA and OW were greatest on herbivores while higher trophic levels demonstrated greater tolerance. However, it is important to note that the combined effects of OA and OW were much lower than their individual effects (OA or OW), with the interaction effect often positioned around zero. This was a somewhat surprising result compared to what experts have predicted

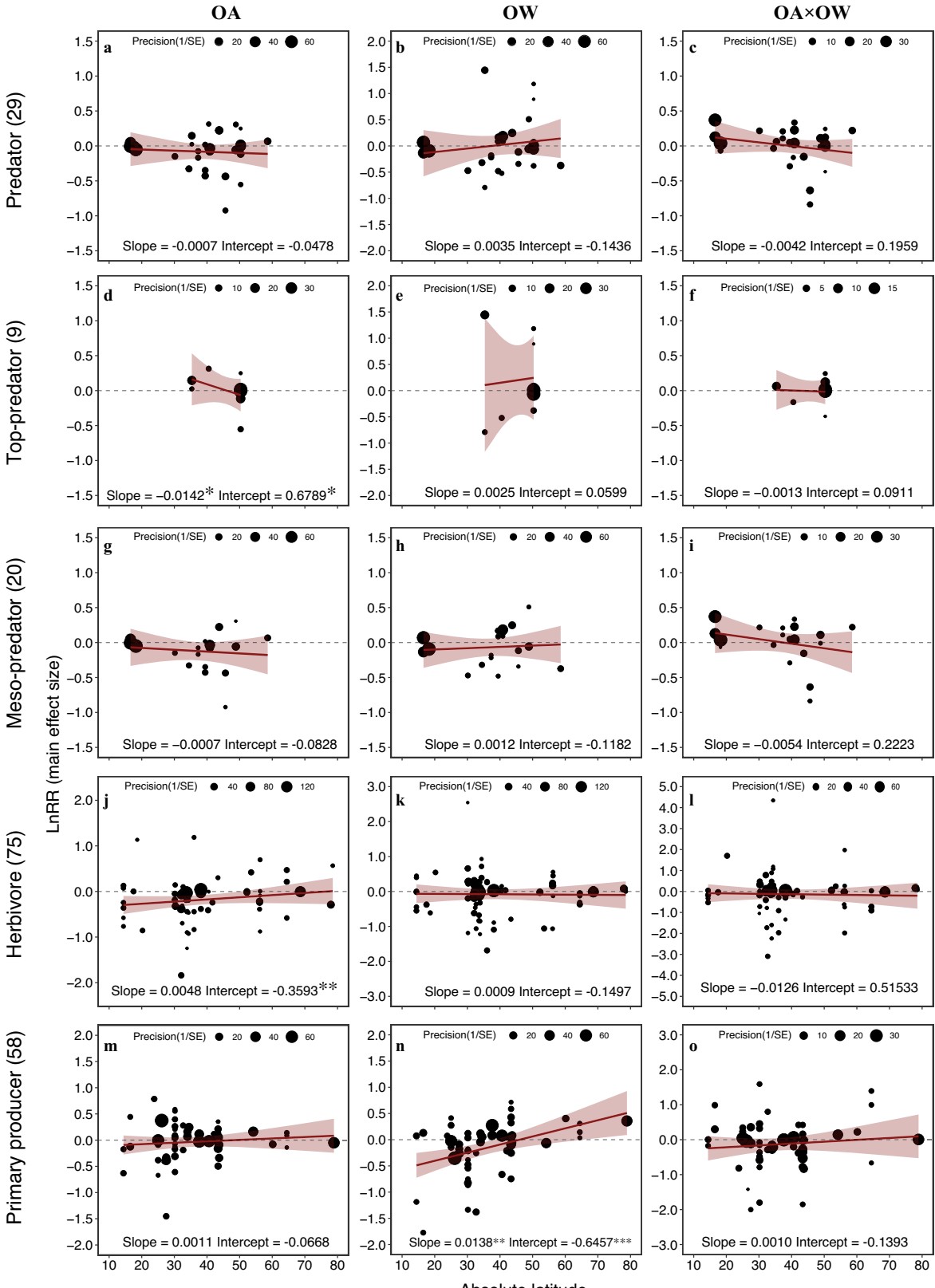

**Fig. 4 | Relationships between the absolute latitude and main effect sizes of stressors.** Latitude-based effects of ocean acidification (**a**, **d**, **g**, **j**, and **m**), ocean warming (**b**, **e**, **h**, **k**, and **n**), and their combined effects (**c**, **f**, **i**, **l**, and **o**) on each trophic level. Slope and intercept were estimated from multi-level meta-regression models included absolute latitude as the moderator. The dark red line shows the mean regression line. Shadow area shows 95% confidence interval. Significant slopes and intercepts were indicated by asterisk ($0.01 < *p < 0.05$; $0.001 < **p < 0.01$; $***p < 0.001$). Numbers between brackets indicate the number of observations. Note the Predator was pooled by the meso-predator and top-predator.

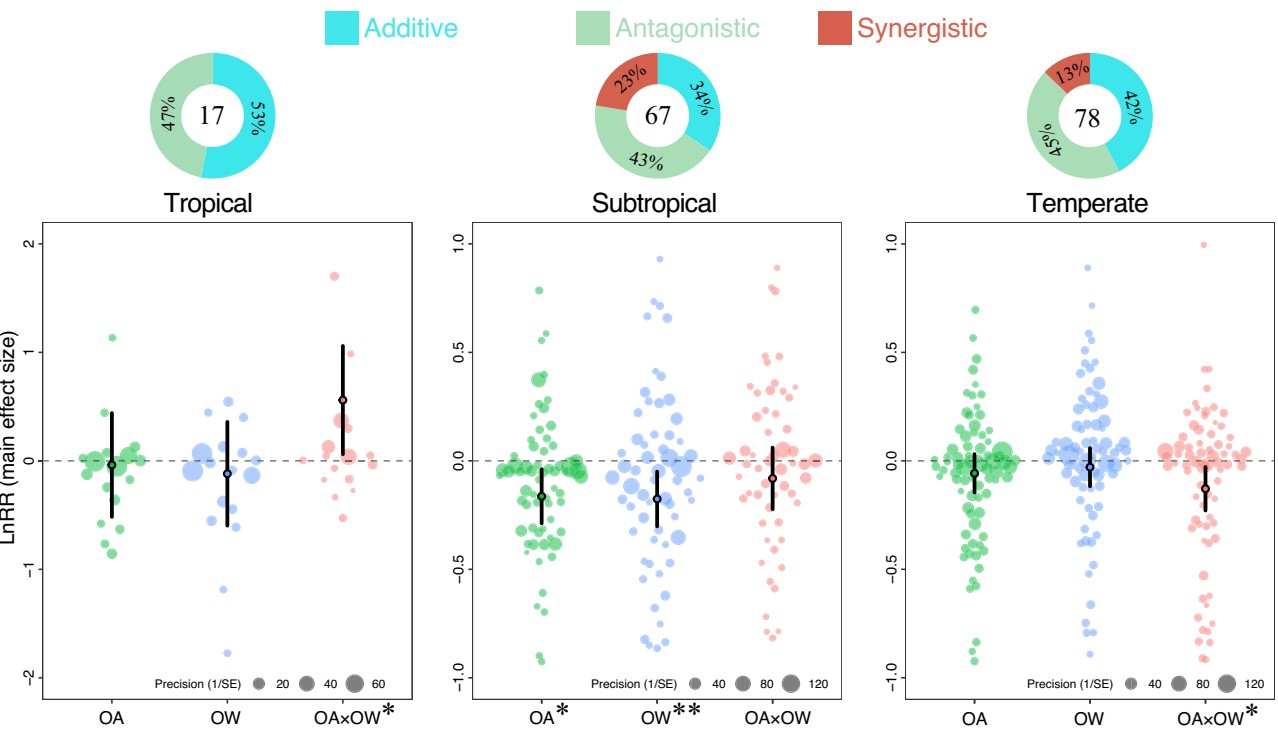

**Fig. 5 | Orchard plots showing mean of main effect size, confidence interval (CIs, bold line), and individual effect size with precision (1/SE) for ocean acidification (green), ocean warming (blue), and their combined effects (red) on climate regions.** Mean effect size and 95% confidence intervals were estimated from multi-level meta-analytic models (two-sided) included stressors as the moderator using main effect sizes. 95% confidence interval does not overlap with zero indicating significant effect showing by asterisk ($0.01 < *p < 0.05$; $0.001 < **p < 0.01$; $***p < 0.001$). Donut charts indicate the frequencies (%) of additive, antagonistic, and synergistic interaction types. Numbers inside donut charts indicate the number of observations ($k$).

about multiple climate stressors and their potential to induce synergistic interactions (see discussion below).

Concerning primary producers, we obtained a somewhat unexpected result in which OA had a positive effect. However, this result may be explained by the fact that high $CO_2$ levels in the surrounding environment can be used as a resource, and have the potential to increase the carbon fixation rates in some photosynthetic primary producers[44–46]. Yet, the combined effect of OA and OW was negative, where the low effect size from the elevated ocean temperature seems to act antagonistically with OA, reversing the individual effects to an even greater negative combined effect. The reason for this result is at this point difficult to disclose, but a similar result was detected for both calcifying and non-calcifying primary producers (Fig. 3). The small effect from OW was also to some extent a surprising result, as related studies have demonstrated that primary producers thrive in high concentrations of $CO_2$ and warm ocean temperatures. Still, this may involve only certain species of primary producers as ref. 47 found that under the "future ocean" regime, larger chain-forming diatoms became dominant at the expense of smaller pennate forms. Such disparities across taxa may be what we detected in Fig. 2, where elevated ocean temperatures (OW) produced to some degree, large variation around the effect size (see blue symbols). In conclusion, our results suggest that there is likely to be interspecific variation in the sensitivity of primary producers to climatic stressors.

Our results also support the hypothesis that calcifying herbivores, such as molluscs and echinoderms, are more sensitive to OA[9,13,48–50]. Interestingly, for herbivores, the combined effects of OA and OW appeared less detrimental compared to their individual effects, and this was irrespective of calcifying or non-calcifying species. The compensatory effect of OW could possibly be explained by the warmer temperatures in OW treatments, which can increase calcium carbonate precipitation kinetics and offset the reduction in calcification caused

by OA, a process observed in e.g. corals[51]. An alternative explanation is that higher water temperatures benefit the development of species (Supplementary Fig. 1), thereby reducing time in the vulnerable planktonic and early benthic juvenile stages that are particularly sensitive to stressors[52–54]. In addition, studies have also suggested that ocean acidification and elevated temperatures impact different metabolic pathways, and in these cases, temperature was the overriding factor, particularly when focusing on mortality[55,56]. Accordingly, herbivores seem more sensitive to OA and OW compared to primary producers when examining individual stressors in isolation. However, the interactive effect on herbivores was less severe, with effect sizes equal to primary producers, and lower compared to the predator groups (Fig. 2), where the main drivers were the combined effects on reproduction and survival (Supplementary Fig. 1).

We found that predators were relatively tolerant to the individual effects of OA and OW[9,41] and also to their combined effects. This suggests that tolerance of predators to one stressor (OA or OW) may confer tolerance to another (OW or OA) stressor when both stressors act on the same physiological or ecological processes, or action pathways of stressors interact[57]. For example, a study in a bony omnivorous fish demonstrated that exposure to an acute sublethal elevated temperature can lead to increased tolerance to acidification challenges[58] due to the linkage between $CO_2$ levels and the expression of the heat shock proteins Hsp70 and Hsp90. The similar magnitude of effects by individual OA, OW, and their combination suggests that co-tolerance may prevail among higher trophic levels in response to multiple climatic stressors[59,60].

Our study revealed trophic differences in response to combined global change stressors, where higher trophic levels seem to be more tolerant to climate stress than lower trophic levels, a pattern which partly has been confirmed for single environmental stressors by two previous studies[9,41]. However, other species-specific trait

characteristics may additionally contribute to marine trophic differences in response to climate change, such as variation in body size[61–63], functional groups within trophic levels[40], differential ability to control body status and physiological processes[64–66], and different activation energy values and metabolic rates among trophic levels[24,67,68]. However, none of these variables are likely to individually drive the trophic differences, instead they may interact together to contribute to the observed variation[69].

Overall, our meta-analysis showed that synergistic interactions of OA and OW are much less common (16%) compared to antagonistic (44%) and additive (40%) interactions on marine species. Our results contradict results from a previous meta-analysis[49], which found that synergistic interactions between OA and OW dominated in marine ecosystems[70]. The disparity between the studies could likely be due to difference in selection criteria of literature, where in our meta-analysis we only included fully factorial experiments. On the other hand, our findings are consistent with other recent reviews[27,31,33,71,72], indicating far fewer synergistic interactions than previously thought. It was also evident that the proportion of synergistic interactions decreased with increasing trophic rank, from primary producers (17%), herbivores (17%), meso-predators (10%), to no synergistic interactions detected among top-predators. The reduction in synergistic interactions, which principally are viewed as detrimental[33], while moving up the food web may further support previous results that higher trophic levels may be less sensitive to climatic stressors than lower trophic levels.

There is a growing interest in how climate change impacts on marine organisms change along latitudinal gradients[3,73]. It is, for example, widely known that biodiversity is more pronounced in the tropics and species richness declines with increasing latitude[73–75], but see ref. 76. The pattern is evident in both terrestrial and marine realms, and is strongly correlated with temperature[77,78]. A study ref. 73 recently showed that species richness has declined around the equator, particularly in latitudinal bands with average annual sea surface temperatures exceeding 20 °C[79]. For elevated ocean temperatures, we confirm the findings of recent studies in that the relationship between absolute latitude and main effect size were all positive in the direction of higher latitudes (Fig. 4). Tropical species may be more sensitive to elevated temperatures since they already live close to the upper limits of their temperature tolerance[3]. What was additionally interesting, was the apparent increase in variation of effect size found at 30° N/S and at 45° N/S, particularly for primary producers and herbivores (Fig. 4). This variation aligns with latitudinal shifts from the tropics through subtropics to the temperate zone. While the causality of ocean warming and acidification effects is well-supported, the influence of latitude, a factor not experimentally controlled, on these effects warrants cautious interpretation. This caution is particularly pertinent as the tropics and subtropics expand poleward, influencing how various marine organisms respond to climate change through mechanisms like rapid adaption, adaptive phenotypic plasticity, or migration capacity[80]. With that said, it is important to remember that the mid-latitudes (between 30° N/S and 60° N/S) incorporate Australia in the southern hemisphere, and North America, as well as Europe in the northern hemisphere, and the simple fact is that most research grants and research projects are concentrated in these specific regions.

Our study also aimed to contribute to the ongoing debate on how marine species from different climate regions, specifically tropical, subtropical, and temperate regions, respond to the simultaneous exposure of OA and OW, and whether interactions are synergistic, antagonistic, or additive. Our findings reveal intriguing and, in some cases, unexpected responses to these combined stressors, exposing the complex interplay between climate change-related factors and their impacts on marine species. We found that temperate marine species are significantly more affected when exposed to both OA and OW together, as opposed to these stressors individually, suggesting a synergistic effect. This contradicts the assumption that warming might alleviate the effects of acidification in cold water environments[81–83]. In this scenario, numerous marine species and particularly calcifying organisms are at risk of decreasing growth, reproduction, and survival rates[84], which could lead to considerable changes in marine coastal ecosystems via habitat loss and habitat simplification[85]. In contrast, tropical marine species were not negatively affected when subjected to each stressor individually, but actually experienced a positive interactive effect when both stressors were present. This result is to some extent unexpected, given the many reports of coral bleaching in tropical regions[86–88]. However, we emphasize that our meta-analysis does not elucidate underlying mechanisms and that the credibility of these results relies on various factors, particularly how well experimental conditions replicated natural ecosystems. Still, this finding suggests that several tropical species may have evolved mechanisms to cope with individual stressors and, when combined, these stressors may have either a reinforcing or a complementary effect. However, it would be important to investigate which specific taxa exhibit this positive interaction and whether this resilience can be generalized across all tropical marine organisms. Lastly, an intriguing aspect of the results was obtained for the subtropical group of marine species. Unlike the temperate group, subtropical species displayed compensatory effects, demonstrating antagonism when exposed to the combination of OA and OW. This implies that the negative effects of these stressors are less severe when they occur concurrently, compared to when they act in isolation. The negative single-stressor effect is notable in contrast to tropical species, which may have evolved adaptations to cope with frequent warming events. Subtropical species, on the other hand may be less prepared for extreme environmental changes. However, understanding the specific mechanisms that underlie the antagonistic effect in subtropical species could be pivotal for developing conservation strategies and identifying potential resilient species in the face of climate change.

It is commonly advocated that conservation actions should focus on local stressors, as global stressors are often difficult to control and manage[89,90]. However, to implement effective mitigation strategies, it is key to identify stressor interactions to understand which stressors to act on, and when and where to intervene for prioritizing conservation actions[89,91,92]. For that reason, we propose that trophic levels will be an informative predictor and ecologically relevant for the future conservation actions, particularly, when species interactions are involved[69]. Many research studies have promoted the need for multiple stressor assessments and where the combined effects of OA and OW will instigate greater negative effects than just the sum of the two (i.e., synergistic effects[25,33]). In this meta-analysis, we examined 486 observations from 162 fully factorial experiments, which should be considered a large investigation with a particularly high statistical power. It is therefore notable to observe that the interaction effect between OA and OW often produced weaker negative effects than single-factor effects, and sometimes, reduced the negative effect, with additive and antagonistic interactions dominating. However, since it appears that trophic levels from other environmental realms[22,40,93] can respond differently to individual stressors, the results presented here may be exclusive to the marine realm and for that reason, multiple stressors deserve additional research and a deeper understanding.

## Methods

### Data selection

We used Web of Science (WoS) as our search engine, using the 'Web of Science Core Collection' databases on 21st of April 2021. We used the following search strings: (ocean acidification OR elevat* CO2 OR climat* change) AND (ocean warming OR global warming OR global* change) AND (multi* stress* OR interact* OR combine* OR synerg*) AND (marine OR ocean OR sea). In addition to our literature survey, we also performed a cross-reference check of our database with the literature used and cited in the previous meta-analysis and reviews

focusing on ocean acidification and/or warming[13,48–50,94,95]. To ensure better reproducibility, we reported detailed information regarding our literature search as a PRISMA[96] (Preferred Reporting Items for Systematic Reviews and Meta-Analyse) statement in Supplementary Fig. 2.

We focused our analyses on studies that simulated realistic future scenarios, typically based on IPCC (AR5) scenario-representative concentration pathways (RCP 8.5). Under these pathways, $pCO_2$ is predicted to increase, on average, from the current levels of ~400 to ~1000 µatm by the end of the century, while sea surface temperature is predicted to rise by an average (±1 SD) of ~3.7 ± 0.7 °C[7,8], compared to average sea surface temperatures in the 1990s.

To be considered, studies had to be fully factorial experiments that incorporate the mean effect of control, individual stressor (OA and OW), and combined effect treatments of OA and OW, as well as the sample size and the standard deviation (or standard error, or 95% confidence intervals). Studies that used acid to manipulate carbonate chemistry were excluded because they did not mimic the expected changes in $HCO_3^-$ concentrations and dissolved $CO_2$. For studies that included stressors other than acidification and warming, we only used the control level (as determined by the author(s)) of the additional stressor[49]. Studies that focused on a combination of direct and indirect impacts (e.g., lower trophic levels response to climate change and predator removal via trophic cascades) were excluded for two reasons: (i) there is difficulty to define the control level of the indirect effect; (ii) before addressing indirect effects, it is necessary to first understand direct effects and their interactions.

In the literature, we focused on the following organismal response variables: calcification, development, growth, metabolism, reproduction, and survival (mortality data were transformed to survival with [1 - mortality])[13,49,95]. For multiple experimental treatments (where more than one treatment group was compared to the control group), we included the highest testing value that was within the range of the RCP 8.5 scenario, for example, if an experiment simultaneously included 2 °C- and 4 °C-increase treatments, the 4 °C-increase treatment was selected. When several measurements were taken for one response variable (i.e., body mass and length for growth), we only used the most inclusive one[13,48]. For studies with time series experiments, we only included the response reported at the end of the experiment. When a single experiment reported several responses related to the same organism (e.g., growth, calcification, and metabolism of the same organism were reported simultaneously), all responses were included. Furthermore, several studies involved multiple species or locations, which were all taken into account. Although the above selection criteria may lead to the risk of non-independences of data, they however ensure that we did not lose important information and statistical power. When available, data were obtained from the data repository platforms online. When not available, data were extracted from tables if possible, or from graphical images in publications using the software GraphClick (v. 3.0; Neuchatel, Switzerland) and WebPlotDigitizer (v.4.4; www.arohatgi.info/WebPlotDigitizer).

## Effect size calculation and interaction type classification

We calculated individual, overall, and interaction effect sizes for each individual observation using ln-transformed response ratio[97] and following methods given by Morris et al.[98]. Response ratios quantify the proportional change in responses resulting from experimental treatments and ln-transformed response ratio is commonly used as it has robust statistical properties and can be easily interpreted. In addition, the response ratio is, on average, a more powerful and less biased effect size than other effect sizes (i.e., Standard mean difference or Standard mean difference with heteroscedasticity) for global change meta-analysis[99].

In factorial experiments, the effect of a stressor can be measured in two ways: by comparing the treatments with and without that stressor in the absence of the other stressor (individual effects) or by comparing the mean performance in the two treatments in which the stressor is present vs. the two treatments in which the stressor is absent (overall effects)[98]. In other words, individual effects reflect the response in the presence of a stressor alone with respect to the control, while overall effects compare the net effect of a stressor in the presence and absence of a second stressor, which provide a more realistic measure of a stressor's effect[50]. We compared individual effect sizes with interaction effect sizes for classification of interaction type and presented main effects of OA and OW as the true effect.

Thus, individual effect sizes of OA ($LnRR_{oa}$) and OW ($LnRR_{ow}$) were calculated relative to the control (CT) as:

$$LnRR_{oa} = \ln\left(\frac{\bar{X}_{oa}}{\bar{X}_{CT}}\right) = \ln\bar{X}_{oa} - \ln\bar{X}_{CT} \tag{1}$$

$$LnRR_{ow} = \ln\left(\frac{\bar{X}_{ow}}{\bar{X}_{CT}}\right) = \ln\bar{X}_{ow} - \ln\bar{X}_{CT} \tag{2}$$

where $\bar{X}_{oa}$, $\bar{X}_{ow}$ and $\bar{X}_{CT}$ are the mean values of the corresponding groups. For each factorial test, the main effect of OA ($LnRR_{OA}$) was calculated as:

$$LnRR_{OA} = \ln\left(\frac{\frac{1}{2}\left(\bar{X}_{OA} + \bar{X}_{OAW}\right)}{\frac{1}{2}\left(\bar{X}_{CT} + \bar{X}_{OW}\right)}\right) = \ln\left(\bar{X}_{OA} + \bar{X}_{OAW}\right) - \ln\left(\bar{X}_{CT} + \bar{X}_{OW}\right) \tag{3}$$

where $\bar{X}_{OA}$ and $\bar{X}_{OW}$ are the mean values of the corresponding groups, and $\bar{X}_{OAW}$ is the mean value of the combined effects. The sampling variance was calculated as:

$$S^2(LnRR_{OA}) = \left(\frac{1}{\bar{X}_{OA} + \bar{X}_{OAW}}\right)^2 \left(\frac{S_{OA}^2}{N_{OA}} + \frac{S_{OAW}^2}{N_{OAW}}\right)$$
$$+ \left(\frac{1}{\bar{X}_{OW} + \bar{X}_{CT}}\right)^2 \left(\frac{S_{OW}^2}{N_{OW}} + \frac{S_{CT}^2}{N_{CT}}\right) \tag{4}$$

where $S_{OA}^2$, $S_{OW}^2$ and $S_{OAW}^2$ are sampling variances of OA, OW, and their interaction groups, $N_{OA}$, $N_{OW}$, and $N_{OAW}$ are sample sizes of the corresponding groups. The overall effect of OW ($LnRR_{OW}$) and its sampling variance ($S^2(LnRR_{OW})$) were obtained by switching $\bar{X}_{OA}$ and $\bar{X}_{OW}$, $N_{OA}$ and $N_{OW}$ and $S_{OA}$ and $S_{OW}$ in Eq. 3 and Eq. 4.

The interaction effect size of OA and OW ($LnRR_{OAW}$) was calculated as:

$$LnRR_{OAW} = \ln\left(\frac{\bar{X}_{OAW}}{\bar{X}_{OW}}\right) - \ln\left(\frac{\bar{X}_{OA}}{\bar{X}_{CT}}\right) = \ln\left(\bar{X}_{OAW}\bar{X}_{CT}\right) - \ln\left(\bar{X}_{OA}\bar{X}_{OW}\right) \tag{5}$$

which has a sampling variance:

$$S^2(LnRR_{OAW}) = \left(\frac{S_{OA}^2}{\bar{X}_{OA}^2 N_{OA}}\right) + \left(\frac{S_{OW}^2}{\bar{X}_{OW}^2 N_{OW}}\right)$$
$$+ \left(\frac{S_{OAW}^2}{\bar{X}_{OAW}^2 N_{OAW}}\right) + \left(\frac{S_{CT}^2}{\bar{X}_{CT}^2 N_{CT}}\right) \tag{6}$$

The interaction effect size for each factorial experiment was calculated by comparing the null predicted additive effect to the actual observed effect of both stressors[27] (Eq. 5; OA and OW in our case). Therefore, each interaction effect size was based on the absolute ln-transformed difference between the observed net effect of dual stressors against a hypothetical additive outcome based on the product of their single independent effects (calculated by Eqs. 1 and 2). Following Jackson et al.[27], we inverted the sign of the interaction effect size when the additive individual effects were negative (i.e., when both

individual effect sizes were negative, or the negative individual effect size had the higher absolute value if they are in opposing[71]; determined by Eqs. 1 and 2). By doing this, we could use interaction effect sizes irrespective of their directionality to classify interaction type for each experiment[71]. This means that a positive interaction effect size represents a synergistic interaction that a combined effect greater[71] (more positive or more negative) than the addition of their single effects[100], and a negative interaction effect size reflects antagonism. An interaction effect size across zero indicates an exact additive effect of two stressors (i.e., their combined effect same as the addition of their individual effect).

### Multi-level meta-analysis models

The publications that included in our dataset often report more than one effect size; for example, they measured different responses (e.g., growth and calcification); they used multiple species or conducted experiments at multiple sites. Furthermore, experiments may be conducted by the same author(s) with species from the same water or country. This can however result in correlations (clustering) among these effect sizes, which invalidates model assumptions of independence[101]. To manage the potential issues of non-independence, we used random effects models with sampling variance-covariance matrices for our meta-analysis.

We first identified the optimal random effects structure of a multi-level meta-analytic model without moderators for the full dataset. We tested random effects including a study identifier (unique identifier per publication), a species identifier (unique identifier per species), while an individual effect size identifier was always included in the model (in order to quantify residual heterogeneity), the country (unique identifier per country) in which the experiment was performed, and the author group that studies shared the first author (Supplementary Table 1). All of the random effects, but the effect size identifier, were possible clustering factors. Model comparisons were conducted on models fitting with Maximum Likelihood[102]. In addition to the optimal random effects structure, variance-covariance matrices that assumed a correlation of 0.5 between effect sizes were calculated to describe the relationship among effect sizes that shared sampling errors (i.e., effect sizes from same organisms)[9,103].

After the determination of the optimal random effects structure and calculation of variance-covariance matrices (which was used the sampling variance calculated by Eqs. 4 and 6), they were included in a series of multi-level meta-analytic models where selected categorical moderators were treated as fixed effects to assess overall effects at each level of each category using main and interaction effect sizes (calculated by Eqs. 3 and 5). Because we were interested in the effects of OA and OW and their combined effects on trophic levels, we ran multi-level meta-analyses treated stressors and trophic levels as moderators. Trophic levels were classified following an earlier work[9] into primary producer, herbivore, meso-predator, and top-predator. However, discontinuous distribution of trophic position was only evident among lower trophic levels (i.e., primary producer and herbivore), whereas, above the herbivore trophic level, food webs are better characterized as a tangled web where trophic position are continuous[104]. For this reason, we merged the "meso-predator" and "top-predator" levels into one level as the "predator". We presented results for both merged and unmerged of meso and top-predators.

Previous meta-analysis on climatic stressors have shown that marine calcifiers and non-calcifiers differ in their sensitivity to global climatic stressors[9,13,50], thus we performed models on each trophic level of calcifiers and non-calcifiers that also included stressors as the moderator. Earlier works have shown that impacts by climate change are particularly striking for the trophic species such as coral-algal symbiosis[105,106]. Therefore, we also tested the relationship between the latitude and effect sizes to investigate the pattern of effect along the latitude across climatic regions. Additionally, we carried out a subset meta-analysis, treating the stressor as the moderator. In this analysis, all trophic levels were pooled for each climatic region: tropical (between 23.5° North and 23.5° South), subtropical (ranging from 23.5° to 35° in both hemispheres), and temperate (extending from 35° to 66.5° in both hemispheres) (Fig. 1). Notably, since the number of observations in the polar region (areas beyond 66.5° latitude in both hemispheres) was too low ($n = 6$) to provide meaningful interpretation, we grouped them with the temperate category.

### Publication bias and sensitivity analysis

Publication bias threatens the validity of results from meta-analysis when a subset of research findings, such as statistically significant results, are more likely to be published[107]. Thus, it is important to test for publication bias. We evaluated bias in two ways: (1) using a graphical tool of funnel plots; and (2) a modified Egger's regression test[99]. Asymmetry funnels plots indicate there is evidence of publication biases. However, the asymmetry may be due to heterogeneity among effect sizes. Consequently, to reduce the impact of high heterogeneity on the shape of funnel plots, we inspected the funnel plot from the multi-level model including the optimal moderators and random effects. Four moderators (Trophic levels, Calcifiers, and Stressors; See Supplementary Table 2) associated with the main results were tested in isolation and combination by comparing the AIC of models. The optimal models then were used for the Egger's regression test for funnel asymmetry by including the standard error of the effect sizes as a moderator[108]. A non-significant sampling error term indicates no asymmetry and little evidence of publication bias. These two approaches deal with non-independence of effect sizes, while the original funnel plots and Egger's regression does not[109,110]. We did not detect a significant effect of sampling error ($p = 0.4338$) on effect sizes for our dataset (Supplementary Fig. 3).

For sensitivity analysis, we provided results from a more conservative model using variance-covariance matrices with a correlation of 0.9 among correlated effect size sample variances (Supplementary Tables 9–13). Results were comparable to the correlation of 0.5 (presented) and none of any statistical significance was reversed except for the relationship between latitude and main effect size of OA for top-predators due to small sample sizes. In addition, metabolism can be ambiguous with respect to organismal performance where increased metabolism (positive effect size) can be either beneficial or stressful to the species[111,112]. For this reason, we changed the sign of the effect size for metabolic responses, and then recalculated the main effects that used for all analyses. The results were quantitatively similar to the results using original metabolism data. All analyses mentioned in this study were conducted in R with the *metafor* (v. 4.3.0) package[113].

### Reporting summary

Further information on research design is available in the Nature Portfolio Reporting Summary linked to this article.

## Data availability

All data used in this study have been deposited on Zenodo digital repository at https://doi.org/10.5281/zenodo.10198734.

## Code availability

The R code used in this study have been deposited on Zenodo digital repository at https://doi.org/10.5281/zenodo.10198752.

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

## Acknowledgements

We thank the Helge Ax:son Johnsons Foundation for financial support. J.H. acknowledges the generous support by the Nippon Foundation of Japan.

## Author contributions

N.H., P.E.B. and J.H. conceived this study together and the project advanced through close collaboration and discussions.

## Competing interests

The authors declare no competing interests.
