## [Peer Review File · Nature Communications]

Responses of Marine Trophic Levels to the Combined Effects of Ocean Acidification and WarmingReviewer #1 (Remarks to the Author):

Thanks for the opportunity to review the paper entitled "Responses of Marine Trophic Levels to the Combined Effects of Ocean Acidification and Warming".

Key points:

This is an interesting manuscript, but I am surprised how similar it is to Hu et al. 2022 "Meta-analysis reveals variance in tolerance to climate change across marine trophic levels" published in Science of the Total Environment <http://dx.doi.org/10.1016/j.scitotenv.2022.154244>

Similarly, the 2022 paper includes ocean acidification, ocean warming, latitude (by latitudinal region), calcifier and non-calcifier, and the same trophic levels of top predator, meso-predator, herbivore and primary producer (without the overall pooled groups or 'predator' pooled groups).

The current manuscript for review now includes the interactive effects of the two stressors (ocean acidification and ocean warming) and degrees latitude (instead of latitudinal region).

The manuscript addresses an important and widely interesting issue. Data are well-researched, presented nicely although the figure quality is low in the review edition and the text size is small. The manuscript provides an important albeit somewhat incremental advance on the Hu et al. 2022 paper.

It is important that the authors acknowledged the compensatory effect of warmer temperatures, and I am glad they did. However, I think this needs to be further unpacked from the perceived 'stressor' of 'ocean warming', and I would encourage that elevated temperatures not be considered (and thus termed) as only a 'stressor' in this manuscript.

For species not living near their thermal maximum (including many temperate species), warmer temperatures are often a major advantage that accelerate developmental processes, thus importantly temperature influences processes, not necessarily 'stresses' processes. In these cases, it is not really surprising that OA plus OW often cancel out through the major benefit of increased rate processes, including development, at warmer temperatures. I think the authors may have the data to explore this concept more with their latitudinal comparisons.

Other points:

Ocean acidification is considered a global change phenomenon, rather than climate change. Suggest to re-focus to global change rather than 'climate change' through the manuscript.

Ocean acidification is not a 'climatic stressor'.

Overall the manuscript is quite well written, although needs some editing for grammar, typos. There are long paragraphs with many components, suggest to split to make clearer.

First 5 references are all 9 years old or older. Update marine effects references and climate scenarios with newer, more relevant references.

L70-71 – global change stressors vary on a diel and seasonal basis in many ecosystems, not just on coral reefs and tide pools.

L214-215 sentence does not make sense and this is an important opening statement about a conclusion from the results. In the next sentence, authors need to state 'OA or OW' in the brackets, not 'OA over OW' and vice versa to avoid confusion.

L225-226 "Our study represents the first meta-analysis to reveal trophic differences in response to multiple climatic stressors," doesn't Hu et al. 2022 find these kinds of differences to different 'stressors'? I think the authors mean 'combined global change drivers' here.

L414 "which we wanted it always included" needs rewording to make sense.

L441 tropical species?

Typographical errors/issues with the text in the reference file for studied used in the analysis need fixing.

Figure text will need increasing, especially Fig 2 and 3 for the text within the graphical plots.

Reviewer #2 (Remarks to the Author):

The present paper is a meta-analysis on the effect of multiple stressor, in particular ocean acidification and ocean warming, on marine organisms. Authors use their analysis to disentangle pattern resulting from the combined action of these stressors analysing different type of interaction effect and highlighting resulting differences between trophic levels. Considering the salient topic, the inclusion of trophic levels and of important performance traits, allowing to project climate change impact from individual to the higher ecological levels, the present work represent an important contribution to the current literature.

The analysis of the data and the models used are clearly reported and I recognize the work done in dealing with non-independence of the data and the selection of the best random effect for the meta-analytic models.

Although I have a very positive opinion about the statistic, I would ask for some clarification regarding literature research, which I think is necessary to clarify in order to recommend the article for publication.

First, authors state "The search was performed with both combined and alone of the terms(...)". Does this mean that you ran two separate research string with and without Boolean operator? Please can you better explain this sentence?

Second, research strings are usually run in two separate research engine, such as ISI (using Web of Science Core Collection) and Scopus, and hits of these two separate research are then checked for duplicate, while in this case multiple database of ISI have been screened. Is there a particular reason for the exclusion of the latter? What are Web of Science's database used in your literature research? Please justify your choice in this search mode and specify in detail the included databases.

Minor issues: please, report the level of statistical significance in figure's captions (main text and supplementary material).

Please, specify the total number of case studies k included in your meta-analysis

Reviewer #3 (Remarks to the Author):

The manuscript presents a meta-analysis of studies investigating multiple stressors in the marine environment. Noteworthy results include how predators are most tolerant to both individual and combined effects of warming and ocean acidification, synergistic interactions are the least common and decreases with trophic level, interactive effects are similar across trophic levels, and there is a latitudinal gradient in responses.

The work is a useful study for people working on climate change and ocean acidification research projects. It makes the point that further research is needed, but this is already very widely known and evident within this field of research.

This is not the first meta-analysis of multiple stressor studies across more than one trophic level, Harvey et al. Ecology & Evolution 2013 did a meta-analysis of the same factors, ocean acidification and warming on multiple trophic levels. The authors need to correct this statement.

The work will be of significance to the field, and is likely to be highly cited. There aren't methods provided in detail in this paper. Very little information on how the authors carried out the meta-analysis is provided, there are just a few lines (85-91) outlining the number of observations, and how the analysis looked at whether differential responses were observed. Figures 2-4 show main effect size and trophic level or plots of latitude, but details are lacking on how they obtained these results. This is something that needs addressing.

We are grateful for the positive review of our current work and for the consensus among all reviewers and the editor regarding the article's significance. We have addressed all points raised by the three reviewers, but with a special focus of having expanded the analysis in accordance with the suggestions made by Reviewer 1, which has significantly enhanced the paper by incorporating the latitudinal dimension of adaptation to both ocean acidification and ocean warming. Below, we have carefully considered all comments from the reviewers and the editor, and we have highlighted our responses in bold.

REVIEWER COMMENTS

Reviewer #1 (Remarks to the Author):

Thanks for the opportunity to review the paper entitled "Responses of Marine Trophic Levels to the Combined Effects of Ocean Acidification and Warming".

Key points:

This is an interesting manuscript, but I am surprised how similar it is to Hu et al. 2022 "Meta-analysis reveals variance in tolerance to climate change across marine trophic levels" published in Science of the Total Environment <http://dx.doi.org/10.1016/j.scitotenv.2022.154244>

Similarly, the 2022 paper includes ocean acidification, ocean warming, latitude (by latitudinal region), calcifier and non-calcifier, and the same trophic levels of top predator, meso-predator, herbivore and primary producer (without the overall pooled groups or 'predator' pooled groups).

The current manuscript for review now includes the interactive effects of the two stressors (ocean acidification and ocean warming) and degrees latitude (instead of latitudinal region).

The manuscript addresses an important and widely interesting issue. Data are well-researched, presented nicely although the figure quality is low in the review edition and the text size is small. The manuscript provides an important albeit somewhat incremental advance on the Hu et al. 2022 paper.

Reply: We thank the reviewer for recognizing the merits of the paper and finding it interesting. Although there are similarities with Hu et al. 2022, these are primarily visual resemblances, as the current paper is based on an entirely new dataset compared to what was reported in Hu et al. 2022. Furthermore, while the article by Hu et al. 2022 presented noteworthy results based on a large dataset of marine studies, it only focused on the singular effects of Ocean Acidification and Ocean Warming and did not include interactive effects. In the natural world, marine ecosystems are subject to the multiple stressors of ocean acidification and warming simultaneously. By studying the combined and interactive effects of these stressors, we can gain a more complete understanding of the actual challenges facing marine organisms at different trophic levels. Further, focusing on the combined effects of stressors allows us to develop a more nuanced understanding of the effects of combined stressors and their complexity (e.g., whether the interactions of stressors are additive or non-additive). Thus, the current analysis builds on the previous

one (Hu et al. 2022) in important ways and with new data. It should also be mentioned that despite using an entirely new data set and focusing on combined, rather than singular stressor effects, it is intriguing and noteworthy to see that the results from this current study demonstrate similar patterns across trophic levels as those identified in Hu et al. 2022.

The new dataset on which we base our analysis includes only original literature that considers the combined effect of ocean acidification and ocean warming. The present manuscript focuses on the interactive effects and further, identifies the types of interactive effects (i.e., whether interactions are additive, synergistic, or antagonistic). Elucidating the prevalence of the different types of interactive effects is timely and highly relevant to the field, as the traditional view of combined stressors favours synergistic effects, largely based on a predisposition among researchers toward this model, rather than being firmly grounded in solid data. However, recently Jackson et al. (2016) demonstrated that synergistic effects were the least frequent among interactive effects of multiple stressors in freshwater systems. Here, we corroborate these results and confirm similar types of interaction effects in the marine environment. This new focus (e.g., compared to Hu et al. 2022) and its implications are discussed and emphasised in the Introduction of the ms already in the original version (line 72-85). However, in light of this review and with the new analysis and the results that emerged from it (see below) we have expanded on these points in the Discussion (line 333-369). We are therefore grateful to the Editor and Reviewer 1 for pushing us to develop this further.

Finally, in response to Reviewer 1's insightful comment about considering an additional analysis focusing on how the interactive effects of OA and OW might differ among species from different climate regions, the ms has been further improved. To the best of our knowledge, this additional analysis is novel and strengthens the manuscript as the new results align well with our originally-presented findings. We are indeed grateful for this suggestion (which we should have thought of ourselves!).

It is important that the authors acknowledged the compensatory effect of warmer temperatures, and I am glad they did. However, I think this needs to be further unpacked from the perceived 'stressor' of 'ocean warming', and I would encourage that elevated temperatures not be considered (and thus termed) as only a 'stressor' in this manuscript. **Reply:** We can only agree with the referee, since "*extreme environmental changes*" can certainly function as positive drivers for some species groups (e.g., certain marine algae thrive in excess of CO₂ and ocean warming). This is also particularly true for combined effects where one driver masks the effect of the other driver (i.e., equivalent to where two drugs have a weaker combined effect than either of the two drugs singly). However, upon re-reading the key literature in this field (e.g., Cote et al. 2016; Jackson et al. 2016; Crain et al 2008; Tekin et al. 2020) we were reminded that they all use the term 'stressor'. Although this may not be optimal, for the sake of consistency, we prefer to retain this terminology, but now explain these issues in the Introduction (line 102-105). We have also expanded the Discussion to make these concerns and biases clear – but again, for purpose of consistency with the literature, have chosen to keep the term 'stressors' with an additional clarification following Cote et al. (2016), which states: "*By stressor, we mean any natural*

or anthropogenic pressure that causes a quantifiable change, whether positive or negative, in biological response (akin to 'driver')". Please note, however, that if the referee and/or the editor feel strongly about this wording, or feel that its use may still risk to obscure the actual magnitude and direction of "***extreme environmental changes***", we are happy to change it throughout the ms.

For species not living near their thermal maximum (including many temperate species), warmer temperatures are often a major advantage that accelerate developmental processes, thus importantly temperature influences processes, not necessarily 'stresses' processes. In these cases, it is not really surprising that OA plus OW often cancel out through the major benefit of increased rate processes, including development, at warmer temperatures. I think the authors may have the data to explore this concept more with their latitudinal comparisons.

Reply: This comment is exceptional, and we thank the reviewer for being so insightful and thinking deeply about the study. We are obviously frustrated we could not see this apparent comparison ourselves. In response, we have provided a new test that allows us to disentangle a long-standing debate between two conflicting models:

- 1. In tropical regions, many species already inhabit environments close to their upper thermal limits due to the inherently hot climate. When heat waves occur, these species and their physiology are rapidly pushed beyond their thermal limits, leading to various issues. Although animals in hot climates often possess enzymes adapted to higher temperatures, these enzymes, like all proteins, begin to degrade at temperatures exceeding 40°C (104 °F). For these reasons, warmer temperatures may only provide benefits to organisms in temperate regions. In these cases, warmer seawater temperatures may compensate for the negative effects of ocean acidification.**
- 2. In this model, tropical species would not experience higher seawater temperatures as stressful, because they are already adapted to withstand extreme warming. This scenario suggests that ocean warming could offset the negative effects of low pH (Ocean Acidification, OA) on tropical species. In contrast, in the temperate region, warmer water alone may be insufficient to counteract the negative effects posed by low pH. Whereas scenario 1 considers absolute temperature values (where 40°C is the threshold for protein breakdown), this scenario considers relative values.**

With our new analysis, a comparison among three different climate regions (Tropical, Subtropical, and Temperate), we obtained results there are of considerable interest to the long-standing debate of model 1 vs model 2 (described above), since we found patterns that confirm model 2. In short, we found significant negative interactive effects of OA and OW on marine organisms in Temperate regions, despite neutral effects of each stressor in isolation. In contrast, we found significant positive interactive effects of OA and OW on marine organisms in Tropical regions, despite neutral effects of each stressor in isolation. However, in Subtropical regions, we found that the independent negative effects of OA and OW were neutralized when combined, consistent with a compensation effect. We have now included these analyses and results in the new version of the manuscript, which also includes a new figure (For Methods see line 545-549; For Results see line 200-215). We also include a discussion of these findings (line 333-369). Note, however, that due to

the insufficient number of studies available in the literature, it was not possible for us to further divide the findings in each geographical zones into trophic levels.

Other points:

Ocean acidification is considered a global change phenomenon, rather than climate change. Suggest to re-focus to global change rather than 'climate change' through the manuscript. Ocean acidification is not a 'climatic stressor'.

Reply: Thank you for this remark. We have however struggled with these two comments, since we do not view ocean acidification as independent of climate change. It is possible we are missing the referee's point here, but we consider both ocean warming and ocean acidification part of climate change, as they are both caused by excess greenhouse gasses from the combustion of fossil fuels and other sources. To find a second opinion, we consulted the latest document from IPCC 2023. In the full version (IPCC 2023) at page 112, both ocean acidification and warming are discussed under the headline and the context of 'Climate change'. Again, and related to your previous comment about 'stressors', we do not have strong objections to it. Our primary concern is ensuring that the text is accessible to readers, and that the article uses commonly-accepted terminology to convey the message clearly. If the editor deems it necessary, we are open to making changes.

Overall the manuscript is quite well written, although needs some editing for grammar, typos. There are long paragraphs with many components, suggest to split to make clearer.

Reply: Thank you for pointing this out. We have carefully re-read the complete manuscript, and have made every attempt to correct for overly-long paragraphs and typos.

First 5 references are all 9 years old or older. Update marine effects references and climate scenarios with newer, more relevant references.

Reply: Thank you, references have now been updated.

L70-71 – global change stressors vary on a diel and seasonal basis in many ecosystems, not just on coral reefs and tide pools.

Reply: Thank you for your comment. We agree that global change stressors affect many ecosystems beyond tide pools and coral reefs, accordingly the statement has been revised and added a few new references (line 106-107).

L214-215 sentence does not make sense, and this is an important opening statement about a conclusion from the results.

Reply: We have to ask the editor about clarification since we simply could not identify the sentence in question.

In the next sentence, authors need to state 'OA or OW' in the brackets, not 'OA over OW' and vice versa to avoid confusion.

Reply: Thank you, that is now changed.

L225-226 "Our study represents the first meta-analysis to reveal trophic differences in response to multiple climatic stressors," doesn't Hu et al. 2022 find these kinds of differences to different 'stressors'? I think the authors mean 'combined global change

drivers' here.

Reply: Thank you. The statement is now rephrased.

L414 "which we wanted it always included" needs rewording to make sense.

Reply: Thank you the statement is now reworded.

L441 tropical species?

Reply: Thank you, the typo is now corrected.

Typographical errors/issues with the text in the reference file for studied used in the analysis need fixing.

Reply: Thank you for pointing this out. We have now corrected any errors we could find in the Supplementary Information file.

Figure text will need increasing, especially Fig 2 and 3 for the text within the graphical plots.

Reply: Thank you. We believe the version that all referees had access to during the review process had lower resolution, as our original files are of high quality. As a solution, we have provided each figure as an individual file in PDF format. The vector graphics in the PDFs should allow for enlargement of the figures without compromising the image quality. We hope this will ensure that all symbols and text in the figures stand out clearly.

Reviewer #2 (Remarks to the Author):

The present paper is a meta-analysis on the effect of multiple stressor, in particular ocean acidification and ocean warming, on marine organisms. Authors use their analysis to disentangle pattern resulting from the combined action of these stressors analysing different type of interaction effect and highlighting resulting differences between trophic levels. Considering the salient topic, the inclusion of trophic levels and of important performance traits, allowing to project climate change impact from individual to the higher ecological levels, the present work represent an important contribution to the current literature.

The analysis of the data and the models used are clearly reported and I recognize the work done in dealing with non-independence of the data and the selection of the best random effect for the meta-analytic models. Although I have a very positive opinion about the statistic, I would ask for some clarification regarding literature research, which I think is necessary to clarify in order to recommend the article for publication. First, authors state "The search was performed with both combined and alone of the terms(...)". Does this mean that you ran two separate research string with and without Boolean operator? Please can you better explain this sentence?

Reply: The search strategy involves a combination of relevant keywords and Boolean operators (AND and OR) to refine the search. To make the search process clearer, we have provided the specific search strings used. In this process however, we identified we had overlooked one string of keywords for multiple stressor studies: (multi* stress* OR interact* OR combine* OR synerg*), which are now included. We believe this addition enhances the reproducibility of our research. Please see line 392-401, thank you.

Second, research strings are usually run in two separate research engine, such as ISI (using Web of Science Core Collection) and Scopus, and hits of these two separate research are then checked for duplicate, while in this case multiple database of ISI have been screened. Is there a particular reason for the exclusion of the latter? What are Web of Science's database used in your literature research? Please justify your choice in this search mode and specify in detail the included databases.

Reply: Thank you for your comment. In this study we only used Web of Science (WoS) due to the restrictions of our university library subscription. The database is the 'Web of Science Core Collection' (line 392-393). This approach is similar to several other influential meta-analyses that have only used WoS databases (Kroeker et al. 2010, Ecology Letters; Nagelkerken & Conell.2015, PNAS; Goldenberg et al. 2018, Nature Climate Change; Siviter et al. 2021, Nature, focusing on multiple stressor effects). In addition, several comparisons have been made among academic literature collections and have concluded that WoS is considered to be the more comprehensive database, with a slight edge over Scopus (e.g. Singh et al. 2023, Journal of Information Science).

Minor issues: please, report the level of statistical significance in figure's captions (main text and supplementary material).

Reply: The significance level has now been included in all figure captions, in both the main text as well as in the SI.

Please, specify the total number of case studies k included in your meta-analysis

Reply: Thank you, k (the total number of case studies) has now been specified.

Reviewer #3

The manuscript presents a meta-analysis of studies investigating multiple stressors in the marine environment. Noteworthy results include how predators are most tolerant to both individual and combined effects of warming and ocean acidification, synergistic interactions are the least common and decreases with trophic level, interactive effects are similar across trophic levels, and there is a latitudinal gradient in responses.

The work is a useful study for people working on climate change and ocean acidification research projects. It makes the point that further research is needed, but this is already very widely known and evident within this field of research.

This is not the first meta-analysis of multiple stressor studies across more than one trophic level, Harvey et al. Ecology & Evolution 2013 did a meta-analysis of the same factors, ocean acidification and warming on multiple trophic levels. The authors need to correct this statement.

Reply: Thank you, this statement has now been removed from the manuscript.

The work will be of significance to the field, and is likely to be highly cited. There aren't methods provided in detail in this paper. Very little information on how the authors carried out the meta-analysis is provided, there are just a few lines (85-91) outlining the number of observations, and how the analysis looked at whether differential responses were observed.

Figures 2-4 show main effect size and trophic level or plots of latitude, but details are lacking on how they obtained these results. This is something that needs addressing.

Reply: We were instructed by the Editor to overlook the comment about very little information about M&M.

Reviewer #1 (Remarks to the Author):

General points:

Thanks to the authors for considering the comments carefully. I'm pleased they have taken onboard the suggestions and added new analyses that have improved the paper. Thanks for confirming this is an entirely new data set used in this manuscript; and yes I agree obviously interactive stressors/drivers are key to understand responses.

I couldn't see in the manuscript where the division for the latitudinal comparisons were made – i.e. which latitudes were the cut off for tropical, subtropical and temperate etc? There can be latitudinal variation in temperature that means a given latitude is warmer or cooler than expected for its latitude (e.g. the warming effect of the Gulf Stream makes higher latitudes warmer than they otherwise would be). So how the authors made the distinction among regions e.g. on latitude (presumably) or perhaps average annual temperature is important to state in the manuscript methods. Perhaps some coloured bands or just some dashes on the side of Fig 1 could show this (if degrees latitude were used), and the specific degrees latitude stated in the methods.

I do still prefer the term driver over stressor, since the word stressor has negative connotations; however, authors can use their preference for their terminology. To follow on this point, use of the term multiple stressors likely lead to the search term "multi* stress*" (L395) and I now wonder if 'driver*' should also be considered for a comprehensive search, as some groups and authors tend to use the word 'driver' instead of 'stressor'. I don't suppose the authors will want to go back to the start with this, so perhaps a comment to consider for future work. I wonder how many more studies may (or may not) have been included.

To clarify a previous point, excess GHG emissions result in climate change and suite of other changes (including indirectly OA) that are collectively known as global change. It is my understanding that originally climate change referred to changes in the Earth's climate system including increasing temperatures. Although ocean acidification was originally not technically considered climate change, given the now common use of the word climate change to also include OA, I understand the authors likely wish to stick with their use of climate change.

Specific points:

Fig 2 and 3 – please move the doughnut charts up so that they are not overlapping with some of the data points on the orchard plots.

Fig 4 – please scale the axis so the data points are not covering the legend at the top (e.g. Fig 4e and 4o) or the legend at the bottom (many subplots) making it confusing to interpret.

Fig 5 – OA, OW and OA&OW should have the same colour coding as Fig 2 and 3 for consistency.

To clarify original comments, original L214-215 read 'We found that predators WAS relatively tolerant to OA and OW' – I think this has been fixed now.

Running title does not quite make sense, I think 'by' should be 'of', and 'response' should maybe be 'responses'.

L55 of references 6-8, only one is an actual climate reference (i.e. IPCC reports or climate modelling studies) and that is a 10-year-old IPCC report. The authors have used different references in L406 for the projected 3.7 deg C increase and these references may be good to include in introduction section when the statement is first made in the manuscript.

L135 suggest to use a 'to' or other word in the CI interval so that there are not 2 dashes together, which can look confusing.

Reviewer #1 (Remarks on code availability):

Please note that when viewing the code using the URL the code webpage identifies the author as 'Hu, Nan', thus invalidating the blind per-review process.

Reviewer #2 (Remarks to the Author):

I believe the answers provided by the authors fully respond to the doubts raised during the review phase. I therefore consider this article acceptable for publication in the journal. Congratulations again to the authors for their work.

REVIEWERS' COMMENTS

Reviewer #1 (Remarks to the Author):

General points:

Thanks to the authors for considering the comments carefully. I'm pleased they have taken onboard the suggestions and added new analyses that have improved the paper. Thanks for confirming this is an entirely new data set used in this manuscript; and yes I agree obviously interactive stressors/drivers are key to understand responses.

Reply: Thank you so much for your positive feedback. We appreciate your acknowledgment of the effort made in revising and enhancing the paper. Once again, we like to send a big thanks to reviewer 1, which has indeed improved the paper with your insightful comments and suggestions. We are truly grateful.

I couldn't see in the manuscript where the division for the latitudinal comparisons were made – i.e. which latitudes were the cut off for tropical, subtropical and temperate etc? There can be latitudinal variation in temperature that means a given latitude is warmer or cooler than expected for its latitude (e.g. the warming effect of the Gulf Stream makes higher latitudes warmer than they otherwise would be). So how the authors made the distinction among regions e.g. on latitude (presumably) or perhaps average annual temperature is important to state in the manuscript methods. Perhaps some coloured bands or just some dashes on the side of Fig 1 could show this (if degrees latitude were used), and the specific degrees latitude stated in the methods.

Reply: Thank you for highlighting this aspect. In this study, we used standard geographic and climatological divisions based on latitudinal bands: tropics (23.5° N - 23.5° S), subtropics (23.5° to ~35° in both hemispheres), and temperate zones (~35° to 66.5°). We acknowledge that these boundaries can vary due to geographical features and climate conditions, like the Gulf Stream effect. We have added dashed lines and points colour to show the distinction among regions.

I do still prefer the term driver over stressor, since the word stressor has negative connotations; however, authors can use their preference for their terminology. To follow on this point, use of the term multiple stressors likely lead to the search term "multi* stress*" (L395) and I now wonder if 'driver*' should also be considered for a comprehensive search, as some groups and authors tend to use the word 'driver' instead of 'stressor'. I don't suppose the authors will want to go back to the start with this, so perhaps a comment to consider for future work. I wonder how many more studies may (or may not) have been included.

Reply: Thank you for your insightful suggestion about using the term 'driver' in addition to 'stressor'. We appreciate the distinction you've drawn between these terms and understand your preference for 'driver' due to its more neutral

connotation. We conducted a comprehensive search using the original keywords supplemented with 'driver'. This additional search yielded approximately 500 extra studies. As the original 6500 studies led to 61 studies included after selection, we may miss 4-5 studies. While 'driver' is not as central as 'stressor' so far to our study, we estimate that this additional search may have potentially missed 2 to 3 relevant studies.

To clarify a previous point, excess GHG emissions result in climate change and suite of other changes (including indirectly OA) that are collectively known as global change. It is my understanding that originally climate change referred to changes in the Earth's climate system including increasing temperatures. Although ocean acidification was originally not technically considered climate change, given the now common use of the word climate change to also include OA, I understand the authors likely wish to stick with their use of climate change.

Reply: Thank you for your understanding and for clarifying the evolving use of the term "climate change."

Specific points:

Fig 2 and 3 – please move the doughnut charts up so that they are not overlapping with some of the data points on the orchard plots.

Reply: Thank you, this has now been changed.

Fig 4 – please scale the axis so the data points are not covering the legend at the top (e.g. Fig 4e and 4o) or the legend at the bottom (many subplots) making it confusing to interpret.

Reply: All sub panels in Fig 4 are now rescaled and in the new version the data points are not covering the legends at both sides, thank you.

Fig 5 – OA, OW and OA&OW should have the same colour coding as Fig 2 and 3 for consistency.

Reply: Thanks for the suggestion. Fig 5 has now been replotted with the same colour set as Fig 2.

To clarify original comments, original L214-215 read 'We found that predators WAS relatively tolerant to OA and OW' – I think this has been fixed now.

Reply: Yes. This has been fixed in the revision.

Running title does not quite make sense, I think 'by' should be 'of', and 'response' should maybe be 'responses'.

Reply: Running title is now reworded as suggested.

L55 of references 6-8, only one is an actual climate reference (i.e. IPCC reports or climate modelling studies) and that is a 10-year-old IPCC report. The authors have used different references in L406 for the projected 3.7 deg C increase and these

references may be good to include in introduction section when the statement is first made in the manuscript.

Reply: The reference of IPCC 2014 is now replaced by the latest IPCC 2023 and the two references refers to 3.7 deg C increase are now moved up to the proper statement.

L135 suggest to use a 'to' or other word in the CI interval so that there are not 2 dashes together, which can look confusing.

Reply: All dashes used in CI intervals are now replaced by 'to'.

Reviewer #1 (Remarks on code availability):

Please note that when viewing the code using the URL the code webpage identifies the author as 'Hu, Nan', thus invalidating the blind per-review process.

Reply: Thank you for pointing this out. We are sorry for the mistake about the code.

Reviewer #2 (Remarks to the Author):

I believe the answers provided by the authors fully respond to the doubts raised during the review phase. I therefore consider this article acceptable for publication in the journal. Congratulations again to the authors for their work.

Reply: Thank you very much for your constructive comments and for acknowledging the efforts made in revising the manuscript. Your insights were instrumental in guiding these improvements. Your engagement with our work is greatly appreciated.